# The tumour suppressor APC promotes HIV-1 assembly via interaction with Gag precursor protein

Kei Miyakawa[1], Mayuko Nishi[1], Satoko Matsunaga[1], Akiko Okayama[2], Masaki Anraku[1], Ayumi Kudoh[1], Hisashi Hirano[2], Hirokazu Kimura[3], Yuko Morikawa[4], Naoki Yamamoto[5], Akira Ono[6] & Akihide Ryo[1]

Diverse cellular proteins and RNAs are tightly regulated in their subcellular localization to exert their local function. Here we report that the tumour suppressor adenomatous polyposis coli protein (APC) directs the localization and assembly of human immunodeficiency virus (HIV)-1 Gag polyprotein at distinct membrane components to enable the efficient production and spread of infectious viral particles. A proteomic analysis and subsequent biomolecular interaction assay reveals that the carboxyl terminus of APC interacts with the matrix region of Gag. Ectopic expression of APC, but not its familial adenomatous polyposis-related truncation mutant, prominently enhances HIV-1 production. Conversely, the depletion of APC leads to a significant decrease in membrane targeting of viral components, resulting in the severe loss of production of infectious virions. Furthermore, APC promotes the directional assembly of viral components at virological synapses, thereby facilitating cell-to-cell viral transmission. These findings reveal an unexpected role of APC in the directional spread of HIV-1.

[1] Department of Microbiology, Yokohama City University School of Medicine, Yokohama 236-0004, Japan. [2] Advanced Medical Research Center, Yokohama City University, Yokohama 236-0004, Japan. [3] Infectious Disease Surveillance Center, National Institute of Infectious Diseases, Tokyo 208-0011, Japan. [4] Kitasato Institute for Life Sciences and Graduate School for Infection Control, Kitasato University, Tokyo 108-8641, Japan. [5] Department of Microbiology, National University of Singapore, Singapore 117597, Singapore. [6] Department of Microbiology and Immunology, University of Michigan Medical School, Ann Arbor, Michigan 48109, USA. Correspondence and requests for materials should be addressed to A.R. (email: aryo@yokohama-cu.ac.jp).

Human immunodeficiency virus (HIV)-1 is the causative agent of acquired immunodeficiency syndrome (AIDS) and has evolved to invade the complex human immune system and utilize the host machinery for the propagation of progeny virus[1,2]. It is well known that the orchestrated participation of viral components and host factors is required for persistent and efficient virus replication. Previous comprehensive genome-wide analyses have revealed hundreds of host proteins to be related to HIV-1 replication[3–5]. However, their functional relevance and the nature of their contribution to HIV-1 propagation in the context of diverse cellular functions, such as cell polarity and cell-to-cell communication remain largely unknown.

In the late stage of the HIV-1 replication cycle, the intracellular trafficking of the viral structural protein Gag (also known as Pr55[Gag]) and viral genomic RNA (vRNA) to the plasma membrane (PM) is a crucial step for the efficient production of infectious virions. The Gag precursor is composed of four functional domains: matrix (MA), capsid (CA, also called p24), nucleocapsid (NC) and p6, and two spacer sequences (Sp2 and Sp1). The MA domain is responsible for the PM targeting of Gag polyprotein. In fact, the hydrophobic myristate anchor at the N terminus of MA can insert into the hydrophobic core of PM. Furthermore, a cationic patch of basic residues on MA forms electrostatic interactions with anionic membrane lipids such as phosphatidylinositol 4,5-bisphosphate $(PI(4,5)P_2)$[6–10]. At the PM, both CA-mediated Gag multimerization and NC-mediated vRNA incorporation can drive viral assembly and production of nascent virions[11]. Live cell imaging analysis has suggested also that Gag could be required for stable association of vRNA with the PM[12,13]. However, it is not fully understood how these viral assembly processes are further regulated during and after Gag–PM binding is completed.

Cellular polarity generates dynamic and spatial patterns both inside and outside of the cell. In terms of virus infection, cell polarity creates a more efficient and dynamic assembly process[14]. Indeed, in polarized cells, viral components are dynamically transported to defined domains and/or structures on the PM, including membrane nanotubes, filopodial bridges or uropods, for efficient assembly and budding[15,16]. These specific membrane structures are generally enriched with actin filaments and can provide the topological spaces for not only the formation of infectious viral particles, but also their deliberate spread with spatial orientation. The most extensively studied among them is the virological synapse (VS), in which the nascent virus is directly passed between two apposed PMs from the infected cell to the neighbouring uninfected cell. It is widely believed that cell-to-cell viral transfer is a major mode of infection in lymphoid tissues[17] and is $10^2$- to $10^3$-fold more efficient for spreading HIV-1 than cell-free infection[18]. Hence, developing an understanding of the host factors that contribute to the targeting of HIV-1 components to the specific site of virus assembly such as the VS might provide an important clue to developing a new anti-retroviral strategy.

In our present study, we demonstrate that the tumour suppressor adenomatous polyposis coli protein (APC) directly binds HIV-1 Gag and regulates the intracellular localization of the viral components for directional HIV-1 assembly. Consequently, APC was found to enhance the VS-mediated cell-to-cell transmission of HIV-1. These findings uncover a previously uncharacterized function of APC in HIV-1 replication and thus provide important new insights into the molecular mechanisms underlying HIV-1–host cell interactions.

## Results

### Identification of APC as a HIV-1 Gag-interacting protein.

Gag is a major component of HIV-1 and plays a crucial role in its assembly. To better understand the host proteins that promote this assembly, we used the tandem affinity purification (TAP) approach[19] to identify HIV-1 Gag-interacting protein(s). We purified the Gag-associated complex from the cell lysates of HEK293 cells expressing HIV-1 Gag fused to a C-terminal TAP tag, which contains an IgG-binding motif and calmodulin-binding motif separated by a tobacco etch virus (TEV) protease cleavage site (Fig. 1a). Gag–TAP-bound proteins were separated by SDS–polyacrylamide gel electrophoresis (PAGE) and visualized by silver staining. Specific bands were then excised and in-gel digested with trypsin. Subsequent mass spectrometry analysis identified the APC protein as a candidate of Gag-binding proteins (Fig. 1b; Supplementary Table 1). These also included angiomotin (AMOT), which is a previously reported Gag-binding factor[20], indicating the validity of our current experimental approach. Glutathione-S-transferase (GST) pull-down and immunoprecipitation analysis further revealed the interaction between Gag and endogenous APC in cells (Fig. 1c,d). Furthermore, this interaction was also found in HIV-1-infected T cells, where Gag protein was immunoprecipated with endogenous APC (Fig. 1e).

We next attempted to determine the binding regions on both proteins that facilitate the Gag–APC interaction. For this purpose we performed in vitro quantitative protein–protein interaction analysis (AlphaScreen; amplified luminescent proximity homogeneous assay)[21,22] using recombinant Gag–GST and APC–FLAG proteins (Fig. 2a). In our assay, we used seven truncated constructs derived from APC, since the full-length APC protein was quite difficult to synthesize using a wheat cell-free system due to its larger size ($>300$ kD). Consequently, our AlphaScreen analysis clearly demonstrated that C-terminal region of APC (APC-CT) could associate with Gag (Fig. 2b). Notably, two C terminally truncated APC mutants (1–1,309 and 1–1,462) found in familial adenomatous polyposis (FAP) patients[23,24], failed to interact with Gag (Fig. 2b). We next investigated the binding domain in Gag, which consists of four major subunits, MA, CA or p24, NC and p6. AlphaScreen analysis demonstrated that APC-CT could preferentially bind MA (Fig. 2c). Taken together, our findings suggested that the APC-CT interacts with the Gag MA domain.

### APC facilitates HIV-1 particle production.

To examine whether APC affects viral particle production through its interaction with Gag, we first transfected HEK293 cells with a HIV-1 molecular clone pNL4-3 (encoding full-length HIV-1$_{NL4-3}$ genome)[25], together with APC expression vectors. We found that wild-type (WT) APC, but not a C terminally truncated FAP–APC (1-1,309), significantly enhanced HIV-1 particle production (Fig. 3a). We performed parallel experiments using SW480 colon carcinoma cells expressing only a C terminally truncated form of APC protein endogenously[26], and found that the exogenous expression of WT APC resulted in the prominent increase in HIV-1 particle production (Fig. 3b). Interestingly, we found that APC-CT could not enhance, but rather reduced viral particle production (Fig. 3c), suggesting that while this region is important for Gag binding (Fig. 2b), it has a functionally dominant-negative effect on HIV-1 production.

We next addressed whether the effects of APC on HIV-1 assembly are due to APC's role in Wnt/β-catenin-dependent signalling. We found that the function of APC in relation to HIV-1 seems not to be caused by inhibition of Wnt/β-catenin signalling, since the overexpression of neither β-catenin nor dominant-negative T-cell transcription factor 4 (TCF4) affected the effects of APC on HIV-1 particle production (Supplementary Fig. 1a–d).

The MA region of Gag is essential for membrane binding of Gag and hence for HIV-1 assembly and budding[27]. Meanwhile,

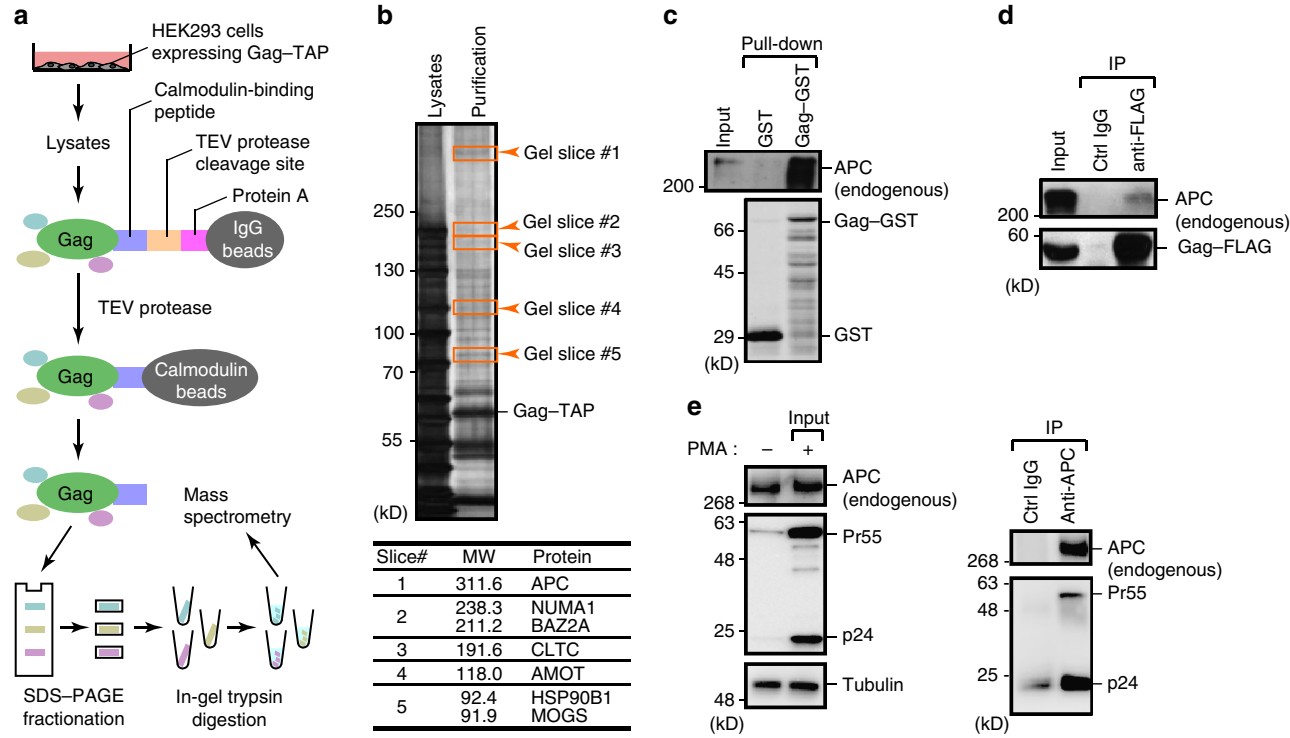

**Figure 1 | Identification of the tumour suppressor APC as a HIV-1 Gag-binding protein.** (a,b) HEK293 cells expressing Gag fused with TAP tag (IgG-TEV-calmodulin-binding motif) were lysed and subjected to TAP-tag-based purification. (a) Proteins bound to Gag–TAP were isolated and detected by SDS–PAGE and silver staining. (b, top) Peptides derived from the excised gel slices (arrowheads) were analysed by matrix-assisted laser desorption ionization–time of flight (MALDI-TOF) mass spectrometry (b, bottom). Detailed information on the proteins identified by mass spectrometry is provided in Supplementary Table 1. MW, molecular weight. (c) Extracts of HEK293 cells were subjected to GST pull-down analysis using either GST or Gag–GST followed by immunoblotting with anti-APC antibody. (d) HEK293 cells expressing Gag–FLAG were lysed and subjected to immunoprecipitation with anti-FLAG antibody followed by immunoblotting with anti-APC antibody. (e) ACH-2 cells stimulated with phorbol 12-myristate 13-acetate (PMA) were lysed and subjected to immunoprecipitation with anti-APC antibody followed by immunoblotting with anti-p24 antibody. Full images for all immunoblots are provided in Supplementary Fig. 4.

the addition of an N-terminal 10-amino-acid Fyn kinase sequence (Fyn(10)), which mediates protein–membrane binding via a triple acylation signal, is sufficient to rescue partially the budding of MA-deficient virus[28]. To assess the impact of the MA domain on the APC-dependent enhancement of virus particle production, we utilized HIV-1$_{NL4-3}$ mutants harbouring either WT Gag or MA-deleted Gag with N-terminal Fyn(10), designated as Fyn(10)WT and Fyn(10)ΔMA, respectively (Fig. 3d), both of which have been shown to have assembly and budding capacity[28]. We observed that the production of virus particles from parental HIV-1 and Fyn(10)WT, but not its MA-deficient derivative Fyn(10)ΔMA, were promoted by APC (Fig. 3d). These results suggested that APC functionally interacts with the MA region of Gag to enhance HIV-1 particle production. Since APC has been found to play a crucial role in a wide variety of cellular processes, including the organization of cytoskeleton networks[29], we next investigated whether the function of APC in promoting HIV-1 particle production is dependent on cytoskeleton dynamics. Parallel analysis with several inhibitors targeting cytoskeletal organization revealed that the inhibition of actin polymerization by cytochalasin B and latrunculin B abrogated the function of APC in facilitating HIV-1 release (Fig. 3e). This suggested that actin dynamics may be a prerequisite for APC-mediated HIV-1 particle production.

**Highly basic region in Gag MA is crucial for APC function.** Previous studies have demonstrated that the highly basic region

(HBR) within the Gag MA has an important role in the targeting of Gag to the specific microdomain of PM for virus assembly[30,31]. To investigate the requirement of these basic residues for the functional interaction with APC, we used a NL4-3 derivative whose basic residues in HBR were substituted by neutral residues (Fyn(10)-Gag(6A2T)-YFP)[30] or with another basic residue (Fyn(10)-Gag(RKswitch)-YFP; ref. 30; Fig. 4a). Despite the changes in HBR, these Gag constructs retained their membrane-binding ability due to the Fyn N-terminal sequence, and a substantial fraction of Gag was found to be present at the PM. Notably, we observed that the production of virus particles from cells infected with HIV-1$_{NL4-3}$ harbouring Fyn(10)WT and RKswitch mutant, but not its 6A2T mutant, was enhanced by APC (Fig. 4a). Consistent with this, AlphaScreen analysis and immunofluorescence staining revealed that APC could interact and co-localize with Fyn(10)WT and RKswitch mutant, but not 6A2T mutant (Fig. 4b,c). These results together suggested that the cationic property on HBR of Gag MA is important for the APC-mediated regulation of HIV-1 assembly.

**APC regulates the recruitment of Gag to the PM.** We next addressed whether APC could modulate the intracellular dynamics of Gag using a C terminally green fluorescent protein (GFP)-fused Gag (Gag–GFP). Consistent with previous reports[32–34], three patterns of Gag localization were observed (Fig. 5a). We analysed for the subcellular localization of Gag–GFP

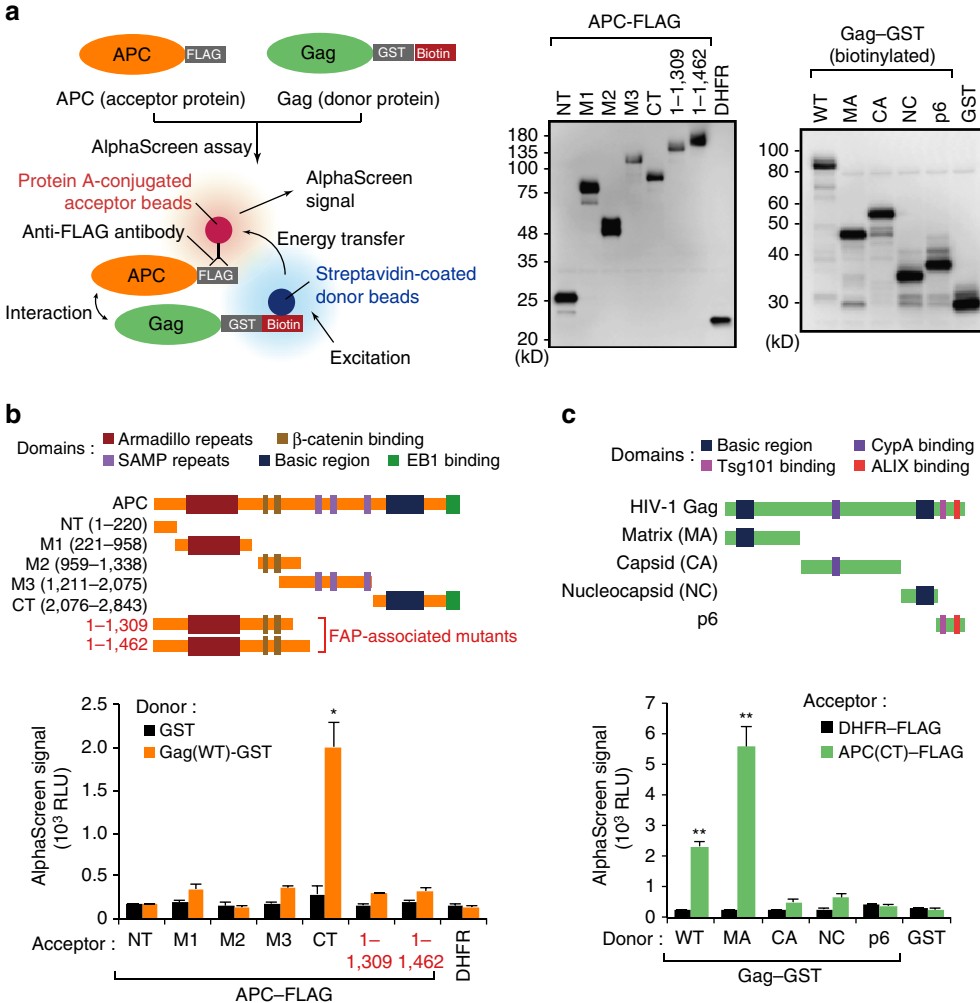

**Figure 2 | The C-terminal region of APC interacts mainly with Gag MA.** (**a**) Schematic representation of the amplified luminescent proximity homogenous assay (AlphaScreen) used to detect Gag–APC interactions. Gag–GST–biotin with donor beads and APC–FLAG with acceptor beads were mixed. If the two proteins can bind to each other, AlphaScreen signals were detected (left). All recombinant proteins used in the AlphaScreen assay were synthesized using a wheat germ cell-free system and detected by immunoblotting with anti-FLAG antibody and horse radish peroxidase-conjugated streptavidin (right). Full images for all immunoblots are provided in Supplementary Fig. 4. (**b,c**) AlphaScreen assays with recombinant APC truncated mutants (**b**) and Gag domains (**c**). FLAG–DHFR (dihydrofolate reductase) and GST were used as negative controls. All graphs are presented as a mean ± s.d. (*n* = 3). \**P* < 0.05, \*\**P* < 0.01, two-tailed unpaired *t*-test.

over 100 cells. At 8 h post transfection, Gag was distributed in a diffuse pattern throughout the cell irrespective of APC expression, whereas its localization was predominantly observed on the PM at 18 h in control cells. Notably, APC-depleted cells exhibited a decreased PM distribution of Gag at 18 h post transfection (Fig. 5a). Following membrane targeting, Gag forms high-order oligomers at the PM[35]. We thus performed direct measurements of Gag–Gag interaction in living cells using nano-bioluminescence resonance energy transfer (NanoBRET)-based protein–protein interaction analysis[36]. Our results demonstrated that APC-depleted cells exhibited less Gag–Gag interactions compared with control cells (Fig. 5b).

Our aforementioned results indicated that APC promotes the multimerization of Gag molecules as a likely result of the enhanced localization of Gag at the PM. To further evaluate the role of endogenous APC in HIV-1 particle production, HCT116 colon carcinoma cells harbouring full-length APC alleles were co-transfected with APC-specific short interfering RNA (siRNA) and HIV-1 molecular clone. The viral production from APC-depleted cells was significantly reduced compared with

the cells transfected with control siRNA, while the total amount of Pr55[Gag] in cells was not altered upon the depletion of APC (Fig. 5c). Subsequently, we investigated the function of APC for multi-cycle HIV-1 replication in T cells. We generated Jurkat T cells expressing APC-specific short hairpin RNA (shRNA), which depleted the endogenous APC protein levels (Fig. 5d). Notably, APC-depleted T cells exhibited a lower level of HIV-1 replication than control T cells (Fig. 5d). This was also observed in primary CD4[+] T cells introduced with APC-shRNA as compared with control cells (Fig. 5e). Taken together, our findings suggest a crucial role of endogenous APC in efficient HIV-1 assembly and spread.

**APC regulates the viral packaging of HIV-1 vRNA.** The accumulation of vRNA at the site of viral assembly and subsequent vRNA packaging into virions are fundamental processes for the production of infectious HIV-1 particles. Several recent reports indicated that the vRNA localization largely depended on Gag association with PM[12,13]. We thus asked

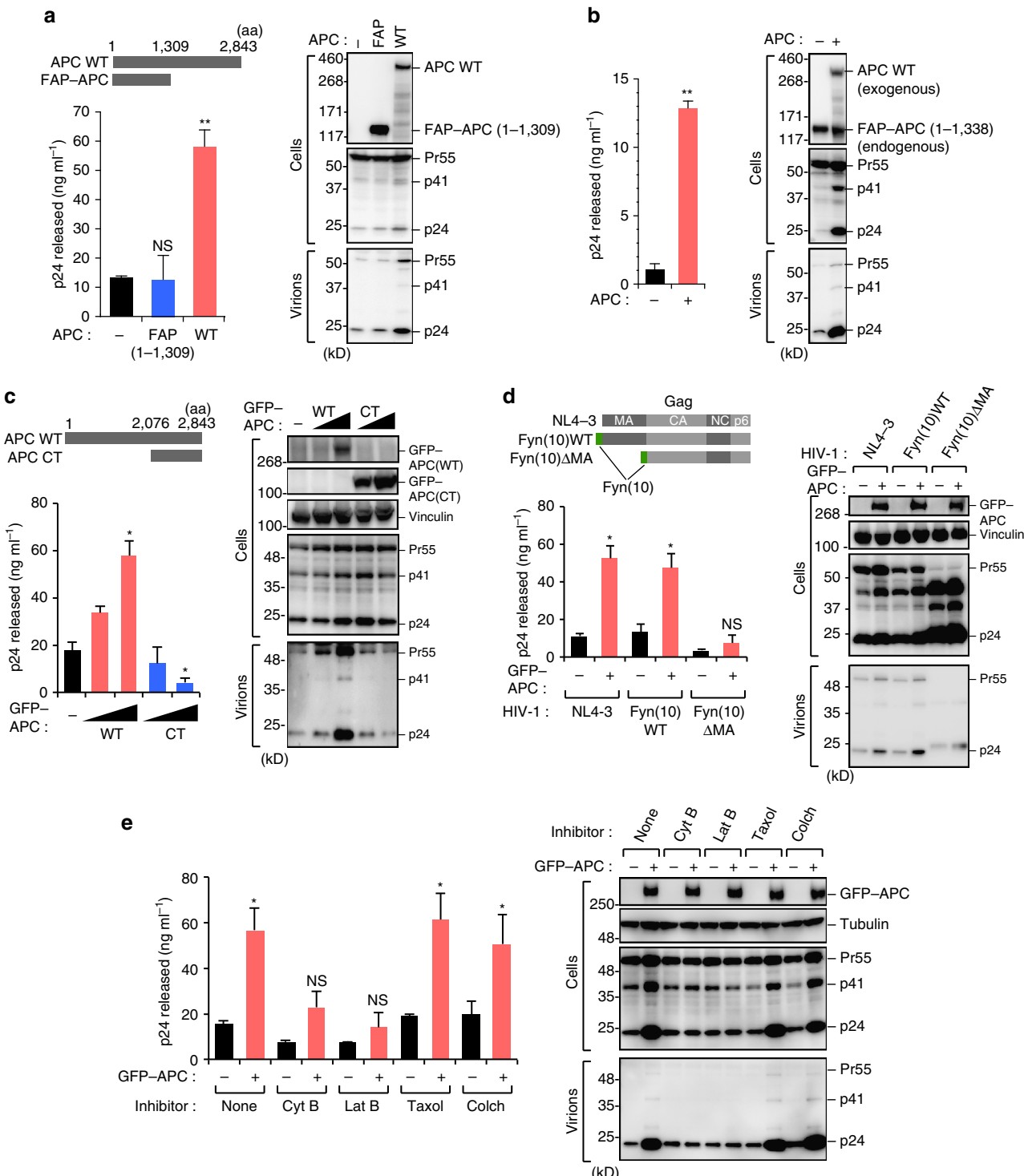

**Figure 3 | APC facilitates HIV-1 particle production.** (**a**,**b**) HEK293 (**a**) and SW480 (**b**) cells were co-transfected with pNL4-3 and plasmids encoding APC (WT or its FAP-related mutant). At 48 h after transfection, cells and supernatants were collected and analysed by immunoblotting using the indicated antibodies. The bar charts indicate the viral p24 antigen levels in the culture supernatants. (**c**) HEK293 cells were co-transfected with pNL4-3 and plasmids encoding APC (WT or its C-terminal domain, CT). At 48 h after transfection, culture supernatants and cell lysates were analysed by p24 enzyme-linked immunosorbent assay (ELISA) and immunoblotting, respectively. (**d**) HEK293 cells were transfected with vectors encoding GFP–APC and the indicated molecular clones encoding WT or MA-deleted Gag. At 48 h following transfection, culture supernatants and cell lysates were analysed by p24 ELISA and immunoblotting. (**e**) HEK293 cells were co-transfected with pNL4-3 and the indicated plasmids encoding APC. The indicated inhibitors were added to cells at 20 h before collecting. At 48 h following transfection, culture supernatants and cell lysates were analysed by p24 ELISA and immunoblotting. The final concentrations of inhibitors were as follows: Cyt B, Cytochalasin B (20 μM); Lat B, Latrunculin B (5 μM); Taxol (100 nM); Colch, Colchicine (50 μg ml$^{-1}$). All graphs are presented as a mean ± s.d. ($n = 3$). NS, not significant; *$P < 0.05$, **$P < 0.01$, two-tailed unpaired $t$-test. Full images for all immunoblots are provided in Supplementary Fig. 4.

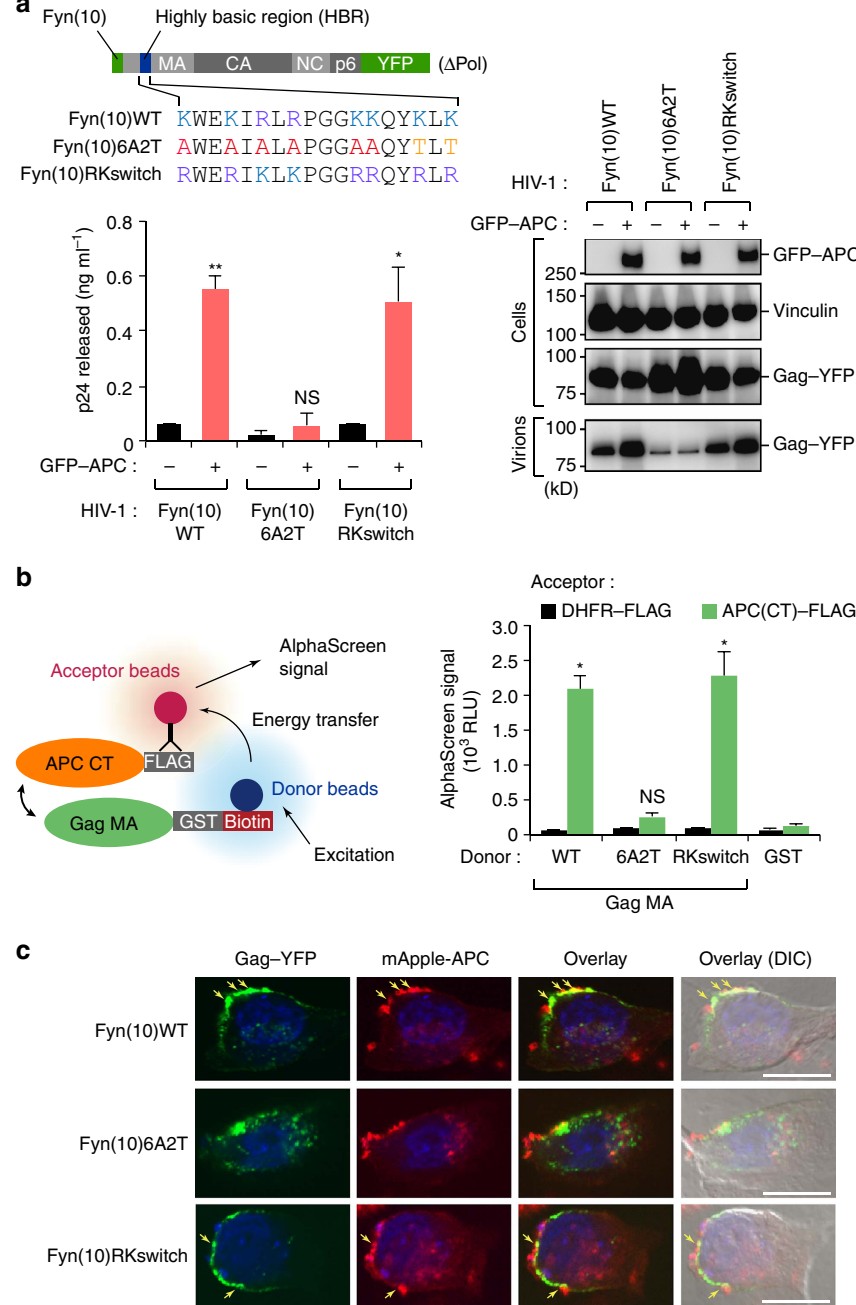

**Figure 4 | The highly basic region in Gag MA is crucial for APC-mediated viral assembly.** (**a**) Schematic representation of the pNL4-3-based derivative whose basic residues in Gag MA HBR were substituted by neutral residues (Fyn(10)-Gag(6A2T)-YFP) or with another basic residue (Fyn(10)-Gag(RKswitch)-YFP). HEK293 cells were transfected with these molecular clones and vectors encoding GFP–APC. At 48 h following transfection, culture supernatants and cell lysates were analysed by p24 enzyme-linked immunosorbent assay and immunoblotting. (**b**) AlphaScreen assays of recombinant APC-CT and MA HBR mutants. DHFR and GST were used as negative controls. (**c**) A549 cells were transfected with indicated molecular clones and together with mApple-APC expression vector. At 24 h following transfection, the cells were fixed and stained with 4,6-diamidino-2-phenylindole (nuclei, blue). Scale bar, 10 μm. The arrows indicate the co-localization of Gag and APC at the plasma membrane. All graphs are presented as a mean ± s.d. ($n = 3$). *$P < 0.05$, **$P < 0.01$, two-tailed unpaired $t$-test. Full images for all immunoblots are provided in Supplementary Fig. 5.

whether APC can facilitate the PM localization and packaging of vRNA. The virions from cell supernatants were prepared and normalized to the Gag p24 levels. We then quantified the amounts of vRNA in the virions and measured their infectivity using TZM-bl reporter cells (Fig. 6a). Interestingly, the levels of virion-incorporated vRNA from APC-depleted cells were significantly reduced (Fig. 6b). Consistently, HIV-1 infectivity was also reduced in the virions from APC-depleted

cells (Fig. 6c). To monitor the intracellular dynamics of vRNA in cells, we created a HIV-1 molecular clone carrying 24 repeats of the MS2-binding site (pNL4-3-pol-MS2x24) in the Pol region (Supplementary Fig. 2a). In this system, co-expression of an RNA-binding protein MS2 fused with GFP allows for the specific tagging of RNA containing MS2-binding sites in cells. As reported previously[12], solely expressed MS2–GFP proteins were mainly observed in the nucleus (Supplementary Fig. 2b).

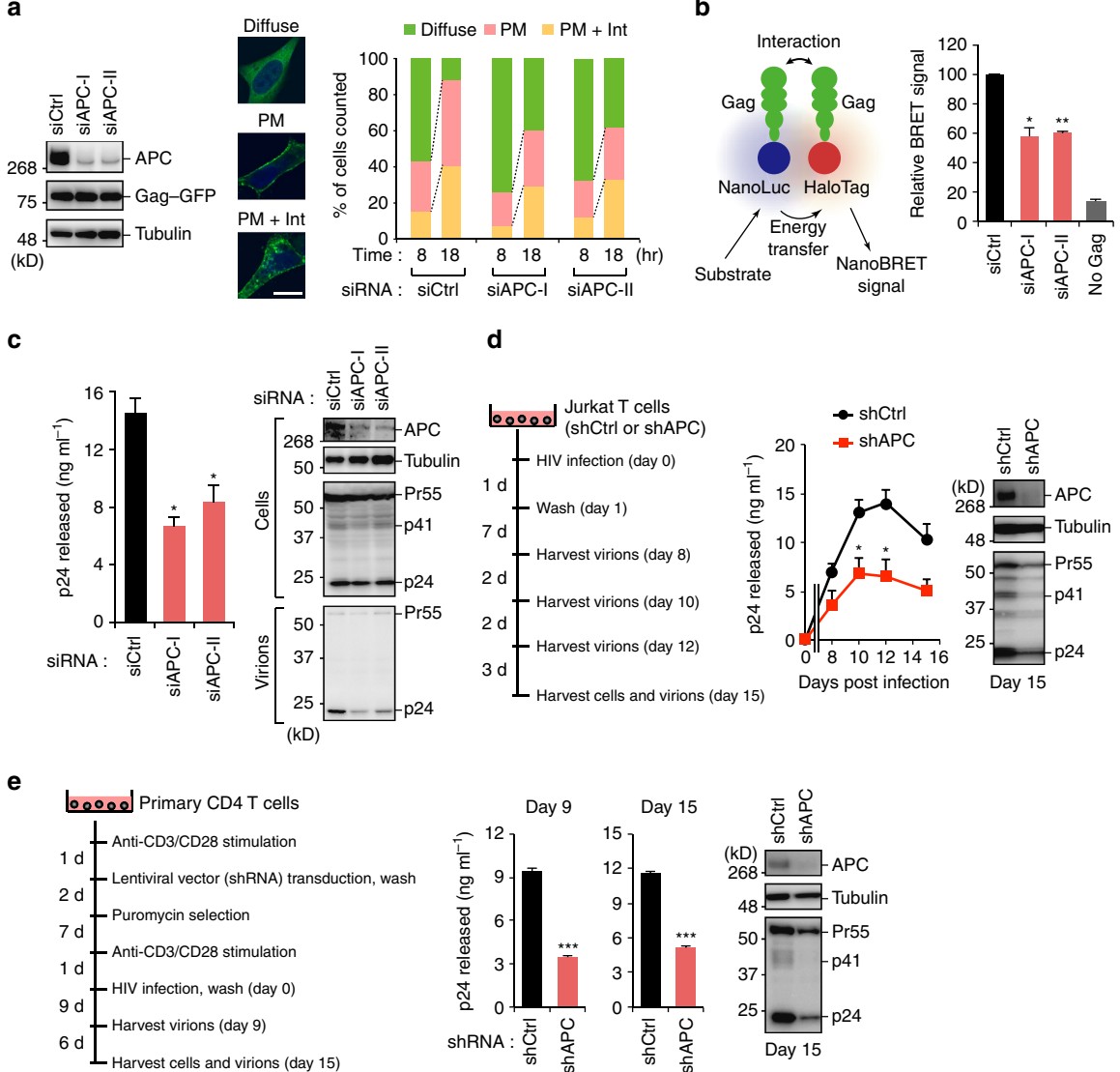

**Figure 5 | APC regulates the recruitment of Gag to the PM.** (**a**) HeLa cells were treated with control siRNA (siCtrl) or two different sequences of APC-targeted siRNAs (siAPC-I or -II) for 24 h before transfection with Gag–GFP expression vector. Blots show the detection of APC, tubulin and Gag–GFP by immunoblotting. At 8 and 18 h after transfection, over 100 cells were analysed for the subcellular localization of Gag–GFP, which was either strongly evident at the plasma membrane only (PM), at the PM and intracellular accumulations (PM + Int), or diffusely in the cytoplasm (diffuse). The data are percentages of the total cells. Scale bar, 10 μm. (**b**) HeLa cells were transduced with control siRNA (siCtrl) or two different sequences of APC-targeted siRNAs (siAPC-I or -II) for 24 h before co-transfection with Gag-HaloTag and Gag-NanoLuc expression vectors. At 48 h after transfection, NanoBRET signals were measured. (**c**) HCT116 cells were treated with either siCtrl or APC-targeted siRNAs (siAPC-I or -II) for 24 h before transfection with pNL4-3. At 48 h after transfection, culture supernatants and cell lysates were analysed by p24 enzyme-linked immunosorbent assay (ELISA) and immunoblotting, respectively. (**d,e**) Multi-cycle viral replication analysis of Jurkat cells (**d**) or primary CD4$^+$ T cells (**e**) stably expressing control shRNA (shCtrl) or APC-targeted shRNA (shAPC). Cells were infected with HIV-1$_{NL4-3}$. Cell supernatants were collected at the indicated time points and subjected to p24 ELISA. The expression of APC and Gag in each cell is also shown. All graphs are presented as a mean ± s.d. (n = 3). *P < 0.05, **P < 0.01, ***P < 0.001, two-tailed unpaired t-test. Full images for all immunoblots are provided in Supplementary Fig. 5.

However, when co-expressed with pNL4-3-pol-MS2x24, these GFP signals were also found in the cytoplasm and at the PM (Supplementary Fig. 2b), indicating that the vRNA had been labelled by MS2–GFP. We analysed for the subcellular localization of vRNA over 20 cells in the presence or absence of APC. By using this system, we could detect a certain level of vRNA at the PM in control cells. Notably, this localization was significantly abolished in APC-depleted cells (Fig. 6d,e; Supplementary Fig. 2c), while the localization of non-viral RNA (luciferase RNA fused with MS2x24 (ref. 37)) was not affected by the APC depletion (Supplementary Fig. 3). These

results demonstrated that APC regulates PM localization of vRNA, resulting in the efficient packaging of vRNA into virions for sustained viral infectivity.

**APC regulates cell-to-cell viral transfer.** We finally assessed if APC is involved in the topology and spread of viral progeny in polarized cells. To this purpose, we investigated whether APC regulates the VS-mediated cell-to-cell viral transfer. In HIV-producing T cells, we found that accumulated Gag, vRNA and also APC at cell-to-cell contact sites that could be highlighted by a VS marker GM1 (ref. 38; Fig. 7a–c). Notably, APC depletion

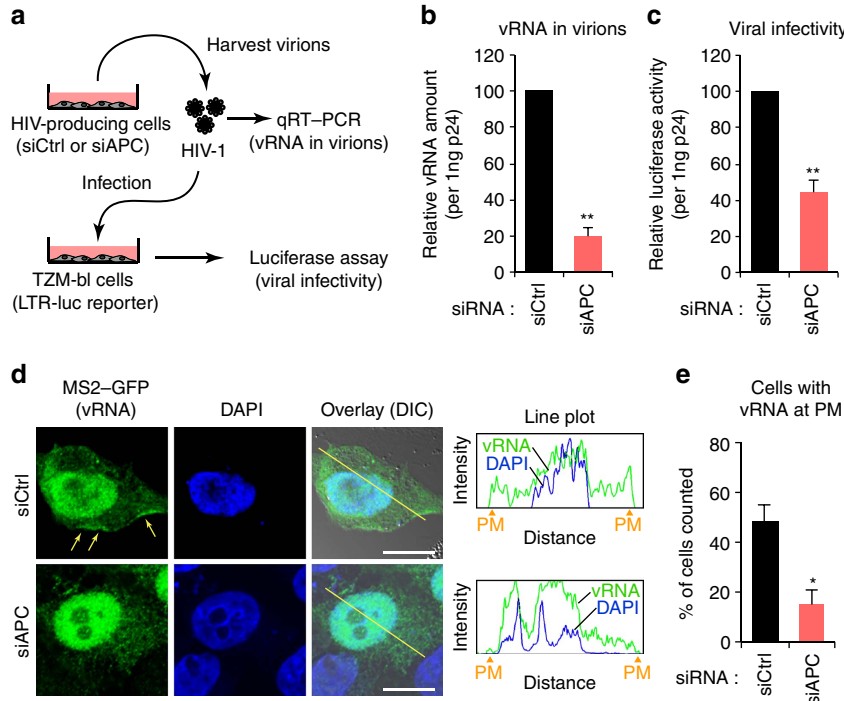

**Figure 6 | APC regulates the PM localization and viral packaging of HIV-1 vRNA. (a)** Schematic representation of virions from HIV-producing cells. HEK293 cells were treated with either control siRNA (siCtrl) or APC-targeted siRNA mix (siAPC) for 24 h before transfection with pNL4-3. After 48 h, virus-containing supernatants were subjected to RNA extraction or used to infect TZM-bl reporter cells to measure vRNA levels or viral infectivity, respectively. **(b,c)** APC depletion reduces the level of vRNA in virions **(b)** and also suppresses viral infectivity **(c)**. Data were normalized to p24 antigen levels in the supernatants. **(d,e)** HeLa cells were treated with either siCtrl or siAPC for 24 h before co-transfection with pNL4-3-pol-MS2x24 and MS2–GFP expression plasmids. At 48 h after transfection, cells were stained with anti-GFP and 4,6-diamidino-2-phenylindole (DAPI) **(d**, left). Scale bar, 10 μm. The arrows indicate the localization of vRNA at the plasma membrane (PM). Line plots in the right panels in **d** indicate the fluorescence intensity with regions of the PM. The bar chart indicates the percentage of cells (at least 20 cells selected from random fields) that vRNA was found at the PM. All graphs are presented as a mean ± s.d. ($n = 3$). $*P < 0.05$, $**P < 0.01$, two-tailed unpaired $t$-test.

significantly reduced the localization of these viral components at cell-to-cell contact sites (Fig. 7a–c). To monitor cell-to-cell HIV-1 transmission, we used a mixed cell culture model. As virus-producing cells, Jurkat T cells were initially transduced with control or APC-targeted shRNA, and then transfected with pNL4-3/Gag–enhanced green fluorescent protein (EGFP), in which GFP was fused to the C terminus of Gag. Uninfected target Jurkat T cells were transiently labelled with Calcein Red-Orange fluorescein dye and then co-cultured with the virus-producing cells in U-bottom plates for 24 h, GFP signals in target cells were detected by flow cytometry to measure cell-to-cell virus transfer (Fig. 7d,e). Using this system, we could quantify the GFP signals that had transferred from producer cells to target cells via VS[39]. We found that the HIV-1 transmission efficiency was prominently reduced when APC was depleted in producer cells (Fig. 7f). Moreover, we performed a cell-to-cell virus transfer experiment using HIV-infected primary CD4[+] T cells. We utilized LuSIV cells[40] as target reporter cells for the detection and quantitation of HIV-1 transfer due to the relatively low virus replication efficiency in primary CD4[+] T cells (Fig. 7g). Our results demonstrated that targeted depletion of APC in HIV-infected primary CD4[+] T cells prominently reduces the virus transfer (Fig. 7h). Taken together, our data suggest that APC regulates cell-to-cell viral transfer in CD4[+] T cells by enhancing the targeting of Gag and vRNA to the VS.

## Discussion

Retroviruses are fully reliant on the host machinery for progeny production and efficient transmission. Accumulating evidence

suggests that the spatial organization of host cells can create a guidance system for dynamic and intensive virion spread[41]. In our current study, we identified the tumour suppressor protein APC as a crucial host factor that orchestrates the assembly and infectious particle production required for directional HIV-1 spread. APC was found to bind and functionally stabilize both Gag and vRNA at specific PM structures such as the VS, resulting in the efficient HIV-1 particle production and spread. Our current study thus unveils a novel role of APC in the topological assembly of HIV-1 that is synchronized with the host cellular architecture and function.

There are two main modes of HIV-1 infection and propagation, both of which are dependent on the host system. Apart from cell-free infection by released viral particles, HIV-1 can spread when nascent viruses are passed directly from an infected cell to a neighbouring uninfected cell. This cell-to-cell infection that arises via the intercellular spatial structure known as a VS is a mode of HIV-1 spread that is several thousand-fold more efficient than cell-free infection[18]. This is therefore a major mode of infection in lymphoid tissues[17,42]. Cell-to-cell infection is likely relevant to the establishment of a viral reservoir[43] and also to the death of CD4[+] T cells induced by HIV-1 (ref. 44). The formation of a VS is required for the interaction between HIV-1 Env (gp120) on infected cells and CD4 on target cells, a process that is accompanied by the rearrangement of actin filaments leading to dynamic changes in cell polarity[17,45]. Hence, the VS-oriented accumulation of viral and cellular components in HIV-1-producing cells should be necessary for efficient cell-to-cell infection. Our current study thus uncovers a distinct function of APC in regulating the directional assembly/budding of HIV-1

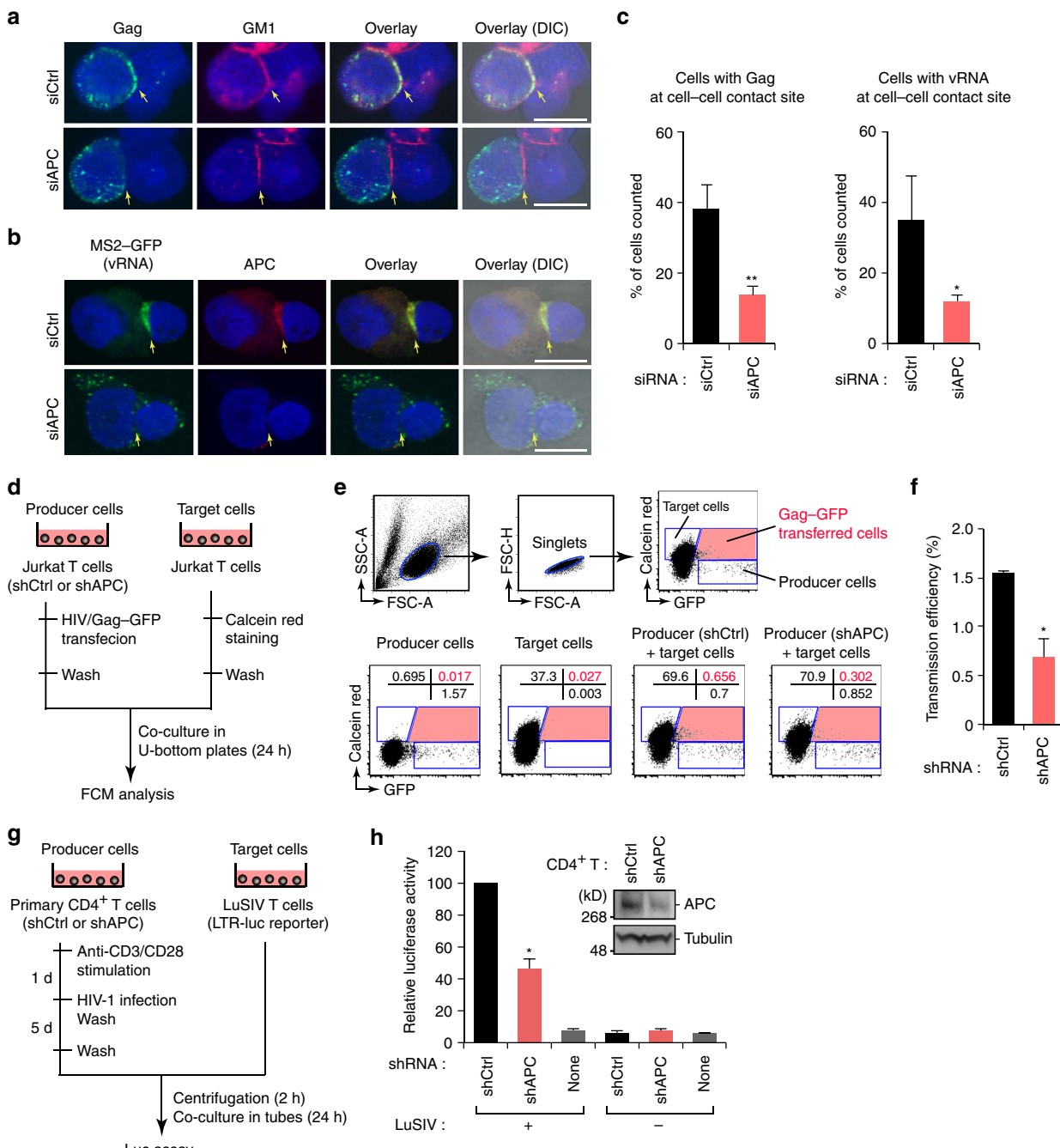

**Figure 7 | APC regulates cell-to-cell viral transfer via virological synapses.** (**a**,**c**) Confocal microscopic analysis of A3.01 cells expressing pNL4-3/Gag–EGFP and either control siRNA (siCtrl) or an APC-specific siRNA mix (siAPC). Cells were stained with Alexa594-conjugated Cholera Toxin Subunit B targeting cell surface GM1 ganglioside (red) and 4,6-diamidino-2-phenylindole (nuclei, blue). Scale bar, 10 μm. The arrows indicate the cell–cell contact site. The bar chart (**c**, left) indicates the percentage of cells (at least 50 cells selected from random fields) that Gag was found at the cell–cell contact sites. (**b**,**c**) Confocal microscopic analysis of A3.01 cells co-expressing pNL4-3-pol-MS2x24, MS2–GFP and either siCtrl or siAPC. Cells were stained with anti-GFP (vRNA, green) and anti-APC (red). Scale bar, 10 μm. The arrows indicate the cell–cell contact site. The bar chart (**c**, right) indicates the percentage of cells (at least 50 cells selected from random fields) that vRNA was found at the cell–cell contact sites. (**d**-**f**) Schematic representation of FACS-based cell-to-cell viral transfer assays used in this study. Jurkat cells stably expressing control shRNA (shCtrl) or APC-targeted shRNA (shAPC) were transfected with pNL4-3/Gag–EGFP, and were used as viral producer cells. Uninfected Jurkat cells stained with Calcein Red-Orange fluorescent dye were used as target cells. Two cell lines were co-cultured in a U-bottom plate for 24 h, and fluorescence signals were then measured by flow cytometry. The transmission efficiency shown in **f** was calculated from the percentage of GFP$^+$ cells among the Calcein-labelled cells by flow cytometry. Data were normalized with the number of virus producer cells. (**g**,**h**) Schematic representation of luciferase-based cell-to-cell viral spread assays used in this study. Primary CD4$^+$ T cells stably expressing shCtrl or shAPC were infected with HIV-1$_{NL4-3}$ (50 ng of p24 antigen) for 5 days, and were used as viral producer cells. LuSIV cells, a reporter cell line for HIV infection with Tat-mediated expression of luciferase, were used as target cells. These cells were co-cultured for 24 h and subjected to luciferase activity (**h**). The expression of APC in cells is also shown. Full images for all immunoblots are provided in Supplementary Fig. 5. All graphs are presented as a mean ± s.d. ($n = 3$). *$P < 0.05$, **$P < 0.01$, two-tailed unpaired $t$-test.

and resultant cell-to-cell viral transmission, that could provide a new mode of therapeutic intervention against HIV-1 infection that targets virus–host interactions.

Another function of APC that has been reported previously is a linking of the intracellular transport and subsequent translation of specific messenger RNAs in intracellular clusters of polarized cells, such as the presynaptic axons of neurons[46]. In retroviral infection, the accumulation of vRNA at the site of assembly is thought to be crucial for its packaging into virions. However, it has remained largely unknown how vRNA is concentrated at virus assembly sites during the late stage of the viral life cycle of HIV-1. In our current study, we have demonstrated a crucial role of APC in the vRNA localization pathway. We found that APC regulates the localization of vRNA at the cell periphery like in VSs where Gag is assembled. Furthermore, we found that APC facilitates vRNA packaging into the virions released in the culture supernatant. We thus hypothesize from our new findings that APC may regulate the localization and functional stabilization of Gag at the VS thereby enhancing its binding to vRNA for its incorporation into HIV-1 virions. Indeed, it has been reported that vRNA can stably localize at the PM in the presence of Gag[12,13]. Another study has also reported that even in the absence of Gag, vRNA can still reach the PM via a diffusion mechanism, but its movement in this case is unstable[47]. We cannot yet completely exclude the possibility that APC directly binds vRNA to enhance its intracellular trafficking, membrane localization and packaging into virions. Further careful analysis will be required to more precisely determine the molecular function of APC with regard to vRNA during HIV-1 particle production.

Gag MA contains a HBR composed mainly of basic amino acids as a bipartite membrane-binding motif[7,8,48,49]. Earlier studies have demonstrated that mutation of HBR causing mislocalization of Gag to intracellular compartments resulted in lower levels of HIV-1 production[48–51]. The binding of the HBR to $PI(4,5)P_2$ has been shown to contribute to PM targeting of the Gag polyprotein[28]. However, $PI(4,5)P_2$ is highly enriched not only at the PM, but also at the rim of caveolae and in coated pits[52]. Recently, host transfer RNA has been found to bind to Gag MA in a manner dependent on the electrostatics of the HBR, while this interaction seems to interfere the PM targeting of Gag[11,53]. Hence, there may be another mechanism of HBR-mediated Gag stabilization at the PM. Otherwise Gag will be missorted to endosomal compartments even after its PM targeting. Our current findings demonstrating that APC prefers to interact with the Gag MA, but not its HBR mutant with neutral amino acids, provide a new concept that the basic amino acids in HBR are relevant to Gag stabilization at the PM, mediated by the function of APC.

The *APC* gene at chromosome 5q21 is transcribed into a nearly 9.5 kb-sized messenger RNA, whose mutations cause FAP and colorectal cancer[54]. The mutation cluster region is located at the central part of the *APC* gene, and the high rates of somatic mutations in this region lead to a premature stop codon, resulting in a nonfunctional truncated APC protein lacking its C-terminal half. In fact, ~80% of FAP patients have truncating mutations in the mutation cluster region of the *APC* gene[55]. APC has a well-recognized function in regulating the Wnt/β-catenin signalling pathway, but its truncated mutants found in FAP patients have completely lost this ability in a dominant-negative manner[56]. Our current analyses suggest that a FAP-related APC mutant does not support HIV-1 production. Conversely, expression of the C-terminal end of APC exhibits a dominant-negative effect upon HIV-1 production, suggesting that the N-terminal and/or middle regions of APC is a prerequisite for its functional regulation of HIV-1. Since human genetic variation is known to affect host susceptibility to HIV-1 infection and disease progression following infection[57], more comprehensive genetic epidemiological studies will be necessary to further explore the biological consequences of APC mutations in HIV-1 infection.

APC plays a central role in suppressing the canonical Wnt signalling pathway that controls cell proliferation and differentiation not only in the intestine[58] but also in the immune system[59]. Previous studies have shown that Wnt signalling is a crucial pathway that regulates many aspects of T-cell biology, including their differentiation and development into effector T cells. In addition to the native function of APC in regulating T cells, we have demonstrated a distinct role of APC in the directional assembly and spread of HIV-1. Interestingly, our current data indicates that Wnt/β-catenin signalling does not directly contribute to the role of APC in HIV-1 production. This unexpected finding evokes the principle that HIV-1 hijacks the host cellular machinery to reproduce itself. However, the dependency of HIV-1 on APC in immune cells could be a promising target for the development of antiviral therapies in the future.

## Methods

**Plasmids.** Human APC genes (UniProt ID #P25054) were amplified from previously described plasmid (pCMV-APC)[60] and subcloned into the pEGFP vector (Clontech, Palo Alto, CA). The mApple-tagged APC was kindly contributed by Dr. Yuko Mimori-Kiyosue (RIKEN, Kobe, Japan). The human-codon-optimized HIV-1 Gag complementary DNA (cDNA) were amplified from previously described plasmid (pGEX-2T-Gag)[61] and subcloned into the pcDNA3 (Thermo Fisher Scientific, Waltham, MA), pCI, pHTC, pNLF1-C (Promega, Madison, WI) and pEGFP (Clontech) vectors. Detailed information regarding the Gag fusion constructs and HIV-1 molecular clones used in the analysis is provided in Supplementary Table 2. For AlphaScreen analysis, APC and Gag cDNAs conjugated with C terminally FLAG and GST-bls (biotin-ligating sequence) tag were subcloned into the pEU-based vector (CellFree Sciences, Ehime, Japan). To visualize vRNA, 24 repeats of the MS2-binding sequence (24xMBS; ACATGAGGATCACCCATGT) were inserted at the EcoRV/AgeI restriction site in the *pol* gene of pNL4-3 carrying the D25N mutation in the protease region. As a non-viral control, a 24xMBS cassette was inserted at the XbaI restriction site at downstream of the stop codon of luciferase gene of pGL4.73 vector (Promega). The expression plasmids for GFP fused to MS2 were obtained from Addgene (#27121).

**Cells and viruses.** HEK293 (#CRL-1573), HeLa (#CCL-2), HCT116 (#CCL-247), A549 (#CCL-185, obtained from American Type Culture Collection; ATCC) and TZM-bl (#8129, from NIH AIDS Reagent Program; ARP) cells were maintained in DMEM supplemented with 10% fetal bovine serum (FBS). SW480 (#CCL-228), Jurkat (#TIB-152, from ATCC) and A3.01 (#166, from ARP) cells were cultured in RPMI containing 10% FBS. ACH-2 (#349) and LuSIV (#5460, from ARP) cells were cultured in RPMI containing 10% FBS, 2 mM sodium pyruvate and 10 mM HEPES. Primary human CD4$^+$ T cells (#2W-200, purchased from Lonza, Basel, Switzerland) were maintained in LGM-3 medium (Lonza) supplemented with 10% FBS. These cells were confirmed to be free of mycoplasma using a MycoAlert mycoplasma detection kit (Lonza). Replication-competent HIV-1 stocks were produced by transient transfection of HEK293 cells with the pNL4-3 proviral plasmid (#114, from ARP)[25]. Culture supernatants containing virus were collected 48 h after transfection and filtered through a 0.45 μm Millex-HV filter (Millipore, Billerica, MA).

**RNA interference and antibodies.** In siRNA experiments, cells were transfected with APC-targeted siRNAs (#HSS100547 and #HSS179353) or control siRNA (#12935300, both from Thermo Fisher Scientific) using Lipofectamine RNAiMAX (Thermo Fisher Scientific) or Hiperfectamine transfection reagent (Qiagen, Venlo, Netherlands) 1 day before DNA transfection or virus infection. Stably APC-silenced cells were generated by transduction with lentiviral particles carrying APC-targeted shRNA (#sc-29703-V) or control shRNA (#sc-108080, both from Santa Cruz Biotechnology, Dallas, TX), and then were selected with 1 μg ml$^{-1}$ (Jurkat cells) and 2.5 μg ml$^{-1}$ (primary CD4$^+$ T cells) puromycin for 7–10 days. Primary CD4$^+$ T cells were stimulated with anti-CD3/CD28 beads (Dynabeads Human T-Activator, Thermo Fisher Scientific) in the presence of 30 U ml$^{-1}$ interleukin-2 (Miltenyi Biotec, Bergisch Gladbach, Germany) 1 day before the transduction of the lentiviral particles. The antibodies used in this study are listed in Supplementary Table 3.

**Proteomic analyses.** HEK293 cells expressing Gag C terminally fused to TAP tag were lysed with IPP150 buffer (10 mM Tris-HCl, pH 8.0, 150 mM NaCl, 0.1%

NP-40) supplemented with 10 mM $Na_3VO_4$, 10 mM NaF and protease inhibitor Complete mini (Roche Diagnostics, Basel, Switzerland). Cell lysates were incubated with IgG sepharose beads (GE Healthcare, Little Chalfont, UK) for 2 h at 4 °C. Beads were washed and incubated with TEV protease (Thermo Fisher Scientific) in TEV cleavage buffer (IPP150 containing 0.5 mM EDTA and 1 mM DTT) for 16 h at 4 °C. The TEV cleavage products was then incubated with calmodulin beads in calmodulin-binding buffer (IPP50 containing 10 mM 2-mercaptoethanol, 1 mM magnesium acetate, 1 mM imidazole and 2 mM calcium chloride) for 1 h at 4 °C. Proteins were eluted with TAP elution buffer (IPP150 containing 10 mM 2-mercaptoethanol, 1 mM magnesium acetate, 1 mM imidazole and 2 mM EGTA) and separated by SDS–PAGE and visualized by silver staining. For in-gel digestion, gel bands were excised from the silver-stained gel, incubated with 50 mM ammonium bicarbonate/60% acetonitrile for 30 min, and then dehydrated. Gel slices were incubated with 12.5 ng $\mu l^{-1}$ trypsin (Promega) in 50 mM ammonium bicarbonate for 16 h at 37 °C. Peptides were eluted from the gel slices with 0.2% formic acid, and then filtrated through a 0.22 μm Ultrafree-MC (Millipore). Mass spectrometric analysis was performed on a LTQ Orbitrap Velos hybrid mass spectrometer (Thermo Fisher Scientific). Protein identification was performed using the Mascot program (Matrix Science, London, UK).

**_In vitro_ protein binding assays and AlphaScreen analysis.** In the GST pull-down assays, GST-tagged Gag was expressed in _Escherichia coli_ BL21 (DE3) cells (#2527I, New England Biolabs, Ipswich, MA) and purified using standard protocols. HEK293 cell lysates were incubated with glutathione beads that had been coupled with Gag–GST proteins. The beads were then washed, and bound proteins were subjected to immunoblotting analysis, as previously described[34,62]. Immunoprecipitation analysis was also performed as previously described[63]. Briefly, cell lysates were immunoprecipitated with 2 μg of antibodies mixed with protein G sepharose (GE Healthcare). In the experiment using ACH-2 cells (HIV-1 latent T-cell clone), cells were stimulated with phorbol 12-myristate 13-acetate (PMA) (100 ng ml[−1]) to reactivate HIV-1 gene expression 2 days before immunoprecipitation. Wheat germ cell-free protein production and AlphaScreen analysis was also performed as described previously[64,65]. Briefly, DNA templates containing FLAG or GST-bls epitope were amplified by PCR with pEU-based vectors and corresponding primers, and then were subjected to the protein production process using the protein synthesizer DTII (CellFree Sciences). AlphaScreen signals from the mixture were detected with an EnVision device (PerkinElmer, Waltham, MA).

**Transfection-based HIV-1 production assays.** Cells in 12-well plates were co-transfected with the HIV molecular clone (100 ng) and either an APC expression vector or empty vector (250 or 500 ng). In the siRNA experiments, cells were transfected with siRNA (20 pmol) 1 day before the transfection of the HIV molecular clone. Two days after transfection, cell lysates and supernatants were collected and subjected to immunoblotting analysis. The p24 antigens in the supernatants were measured with an ELISA kit (Zepto Metrix, Buffalo, NY). To calculate viral infectivity, TZM-bl reporter cells were infected with normalized viruses (1 or 5 ng of p24 antigen) and the HIV-1 long terminal repeat (LTR)-driven luciferase activity was assayed at 2 days post infection. For vRNA detection, total RNA was extracted with TRIzol (Thermo Fisher Scientific) and further treated with DNase I (Takara Bio, Shiga, Japan). After phenol:chloroform extraction and subsequent ethanol precipitation, cDNA was generated with ReverTra Ace (Toyobo, Osaka, Japan) and an oligo(dT)$_{20}$ primer, and quantitative PCR was then performed using SYBR premix Taq (Takara Bio). The PCR primers were as follows: 5′-CATGTTTTCAGCATTATCAGAAGGA-3′ (vRNA-Gag6-Fwd) and 5′-TGCTT GATGTCCCCCCACT-3′ (vRNA-Gag84-Rev)[66].

**Multi-cycle HIV-1 replication assays.** Jurkat (10$^5$ cells) or primary CD4$^+$ T cells (10$^6$ cells) stably expressing either shCtrl or shAPC were infected with HIV-1$_{NL4-3}$ (25 ng of p24 antigen). The latter cells were stimulated with anti-CD3/CD28 beads in the presence of 30 U ml[−1] interleukin-2 1 day before HIV-1 infection, as described above. The infection was performed by centrifugation for 90 min at 500 g in the presence of polybrene (5 μg ml[−1] ●), and cells were then washed to remove the input viruses. Nascent virions produced from infected cells were collected periodically and the p24 levels were measured as described above.

**Microscopic analysis.** For immunofluorescence, HeLa or A549 cells were seeded onto glass cover slips 1 day before transfection. At 24 or 48 h post transfection, the cells were fixed with 4% paraformaldehyde (PFA) and permeabilized with 0.5% Triton X-100. The cells were then stained with primary antibodies and Alexa Fluor-conjugated secondary antibodies. In the experiments using A3.01 T cells, transfected cells were cultured in round-bottom plates for 48 h. Cells were then fixed with 4% PFA and permeabilized with 0.2% saponin (except for surface GM1 staining), and stained with antibodies. For GM1 staining, fixed cells were stained with Alexafluor594-conjugated Cholera Toxin Subunit B (Thermo Fisher Scientific) for 30 min. Microscopic imaging was performed with an FV1000-D confocal laser scanning microscope (Olympus, Tokyo, Japan). Line plots of the fluorescence intensity were generated by ImageJ software (NIH, Bethesda, MD).

Quantitative analysis of subcellular localization of Gag–GFP was performed as previously reported[34,67,68]. Briefly, HeLa cells expressing Gag–GFP were fixed and 20 random fields were inspected. Over 100 cells were analysed for the subcellular localization of Gag–GFP, which was either strongly evident at the PM only, at the PM with intracellular accumulations, or diffusely in the cytoplasm.

**Cell-to-cell viral transfer assays.** For fluorescence-activated cell sorting-based analysis, Jurkat cells stably expressing either shCtrl or shAPC (10$^5$ cells) were transfected with 5 μg of pNL4-3/Gag–EGFP[67] using the Neon transfection system (Thermo Fisher Scientific) in accordance with the manufacturer's protocol. Uninfected Jurkat cells (10$^6$ cells) were labelled with 100 nM of fluorescein dye (CellTrace Calcein Red-Orange AM, Thermo Fisher Scientific) for 20 min at 37 °C, and were used as target cells. These cells were mixed and centrifuged at 500 g for 2 h, and then cultured in U-bottom 96-well plates for 24 h. Cells were fixed with 4% formaldehyde and analysed using a FACSCanto II instrument (BD Biosciences, San Jose, CA). Data were analysed with FlowJo software (Treestar, Ashland, OR). We excluded cell–cell doublets by detecting disproportions between forward scatter (FSC)-A versus FSC-H to analyse the exact efficiency of viral transfer. The transmission efficiency was calculated from the percentage of GFP$^+$ cells among the Calcein-labelled cells. Data were normalized with the number of virus producer cells.

For luciferase-based analysis, primary CD4$^+$ T cells stably expressing either shCtrl or shAPC (10$^5$ cells) were infected with HIV-1$_{NL4-3}$ (50 ng of p24 antigen) for 2 h. Five days later, these cells were mixed with a same number of LuSIV cells (a human T-cell line containing LTR-driven luciferase gene) and then centrifuged at 500 g for 2 h. After co-culturing in a 1.5- ml tube for 24 h at 37 °C, the cells were assayed for luciferase activity.

**NanoBRET assays.** Cells in 96-well plates were transfected with Gag-NanoLuc and Gag-HaloTag expression vectors at a ratio of 1:100. At 48 h post transfection, NanoBRET activity was measured using the NanoBRET Nano-Glo Detection System (Promega). If both proteins are within 100 Å, NanoBRET signals are detected[36].

**Statistical analysis.** The values in all graphs are presented as a mean and s.d. The statistical significance of differences between two groups was tested using a two-tailed unpaired $t$-test with Prism 6 software (GraphPad, San Diego, CA). A $P$ value of $< 0.05$ was considered statistically significant.

**Data availability.** All relevant data are available from the authors on request.

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

## Acknowledgements

We thank Drs Yuko Mimori-Kiyosue, Hiroyuki Gatanaga and Tatsuya Sawasaki for reagents and discussions and Mayu Miyamoto, Noriko Ikawa, Takashi Masaoka, Haruka Sato and Mina Dairaku for their technical support. We also thank Kilian Perrem for proofreading. This work was supported in part by a Creation of Innovation Centers for

Advanced Interdisciplinary Research Areas Program (to A.R.), an NIH R01 grant AI 071727 (to A. Ono), JSPS Grants-in-Aid for Scientific Research 16H05198 (to A.R.) and 16K08814 (to K.M.), and by AMED Grants-in-Aid for Research Program on HIV/AIDS (to A.R. and K.M.).

## Author contributions

K.M. designed and performed the research, analysed the data and wrote the manuscript; M.N., S.M., A. Okayama, M.A. and A.K. performed the research and analysed the data; H.H., Y.M., and A. Ono contributed reagents and analysed the data; H.K. and N.Y. designed the research and analysed the data; and A.R. designed and supervised the research, analysed the data, and wrote the manuscript.

## Additional information

**Competing financial interests:** The authors declare no competing financial interests.

