## [Peer Review File · Nature Communications]

Reviewers' comments:

Reviewer #1 (Remarks to the Author):

Nature Communications manuscript number NCOMMS-16-15882, by Miyakawa et al. presents an intriguing analysis of the effects of the tumor suppressor APC on HIV-1 assembly and infection. The strongest aspect of the manuscript is the characterization of the MA-APC binding interaction, and the analyses of the effects of APC depletion and expression on HIV-1 virus particle release. To me, the virological synapse (VS) data are less convincing to the extent that they detract from the presentation. A revised version of the manuscript should be of interest to HIV researchers. Suggestions for revisions are as follows:

1. All constructs: Readers should not be forced to track down construct details in half a dozen papers. Please include details of exactly where all Gag fusion constructs are linked to their tags.
2. Figure 1: The authors should include a table of all Gag-interacting proteins identified by MS and their spectral count numbers, as well as what proteins were identified in control TAP tag pulldowns. This will be of interest to readers.
3. Figure 2: Did the authors use full-length APC fusions? If not, please explain. Also, based on the cartoon in panel A, it looks as though the GST tag is fused to the N-terminus of MA, which might be expected to perturb its folding. Please address this issue.
4. Figure 3: In panels a-d please show the virion immunoblots and include the Gag CA (p24), p41 and Pr55 bands on the blots. This is necessary to validate release values. Also, with the SW480 blot (panel b), the cellular Gag bands for the APC- lane are much less bright than for the APC+ lanes, which could readily explain for the lack of Gag protein release as virions. Please use a better gel. I would also note that the top of the APC- blot in panel b looks highly contrasted. Perhaps a better blot here would be in order too.
5. Figure 4: Panel A: I do not see a significant difference between the examples of diffuse and PM staining. Better examples should be used. The authors also should use less subjective measurements of diffuse, PM and PM+Int staining. Panel B: I don't find the FRET data convincing. Panel C: Please show the entire virion blot. Panel D: Did the authors score for p24 levels at later timepoints? If so, what were the results?
6. Figure 5: Assuming that "APC targets cellular mRNA to actin-rich plasma membrane protrusions" (page 9), the vRNA localization is likely to be independent of the cis-active vRNA control elements. A control 24xMS2 RNA expressed from a non-retroviral plasmid ought to be tested. It also would be useful to graph cumulative RNA (MS2-GFP) line plots (summed from 20 or so images of each type) for quantification purposes.
6. Figure 6-7: I think the data from these VS model are not so convincing, and that the figures and their discussion can be deleted.
7. Supplementary Figure 3: An analyses of Fyn(10)WT, Fyn(10)6A2T and Fyn(10)RKswitch does contribute to the manuscript, but instead of using the alphascreen, and colocalization studies, the authors should simply perform parallel studies to Figure 3a and include the results as a figure in the paper itself.
8. Discussion: Assuming Figures 6-7 are deleted, the Discussion should be re-written to reflect this.

Reviewer #2 (Remarks to the Author):

The manuscript by Miyakawa et al identifies and defines the role of the tumor suppressor APC in coordinating polarized assembly of HIV. Identified from an affinity purification/mass spectrometry approach they determine that the protein can interact with the MA domain of the Gag. Knockdown studies indicate that it facilitates particle production and recruitment of Gag to the plasma membrane. Interestingly, they also provide some data to suggest that the recruitment of viral RNA to nascent virus particles is regulated by APC. These studies are well performed and supportive of their overall hypothesis, the cell types that are best studied in this manuscript are predominantly epithelial cell lines, and at the very end, an interesting experiment with a T cell line is presented that suggested that APC can play a role in efficient cell-to-cell transfer across virological synapses formed between T cell lines. The experiments performed are generally technically sound and mostly well controlled. The manuscript would be strengthened by studies to show whether APC is expressed in primary T cells, and whether knock down or knock out of APC in primary T cells has an effect on infection and cell-to-cell spread. This study implicates APC as an important new host factor that participates in the polarized localization of viral assembly to the site of cell-cell contact during virological synapse formation.

Specific Comments:

1. Would be important to evaluate the levels of endogenous APC in primary T cells and determine whether this interaction occurs within infected primary T cells.
2. Fig. 7: It is not clear whether the analysis is rigorously excluding cell-cell doublets. The study may also benefit from making stable cell lines that are expressing shRNA or have a deleted APC gene with CRISPR. It would be convincing to show that the levels of APC protein are lower in the presence of siAPC vs siCtrl.

Reviewer #3 (Remarks to the Author):

Through gain and loss of function experiments it is shown here that APC is involved in the assembly and release of HIV particles from infected cells. APC binds to the gag protein of HIV-1 and is essential for release of virions (Fig.3,4), self-association of gag proteins (Fig. 4) and localization of gag proteins and viral RNA at the plasma membrane and virological synapses (Fig. 6) as well as viral transfer to non-infected cells (Fig. 7).

As a non-virologist I consider the experimental strategy and results convincing. They establish APC as a key host-factor for HIV-1 propagation which has obvious consequences to our understanding of HIV-1 biology and will stimulate research into the molecular biology and consequences of APC action.

Points to consider:

- 1 It is formally possible that effects of APC gain or loss of function experiments on HIV assembly are due to APC's role in Wnt signaling, i.e. that they are mediated by β -catenin

dependent signaling rather than or in addition to its interaction with gag. In such a scenario activation of Wnt signaling by APC knockdown or in APC mutated colorectal cancer cells would prevent HIV assembly while inhibition of wnt signaling by full-size APC would stimulate it. It should be tested whether simultaneous knockdown of beta-catenin or expression of dominant-negative TCF blocks the consequences of APC knockdown in some of the key experiments shown in Figs. 4-7. Similarly, effects of b-catenin transfection or treatment of cells with Wnt3a should be determined.

2 Please explain briefly what the pNL4-3 vector encodes for.

3 What is the purpose of the Fyn part in fig. 3d?

4. Fig. 4c and d would fit better with Figure 3

5 Fig. 6 This a collection of neat experiments. However, how can authors state that GAG, vRNA and APC are enriched in protrusion if one doesn't see the rest of the cell? Along that line, is it possible that absence of Gag and vRNA at protrusion after APC knockdown simply reflects lack of membrane association as shown for vRNA in fig. 5?

Responses to the comments of Reviewer #1

We are grateful for the insightful and constructive suggestions by this reviewer. Reviewer#1 noted some of the intriguing effects of the tumor suppressor APC on HIV-1 assembly and infection. Furthermore, this referee also accepted the significance of the MA-APC binding interaction and the effects of APC depletion and expression on HIV-1 virus particle release. However, this reviewer thinks that the virological synapse (VS) data are less convincing to the extent that they detract from the presentation. We have responded to the concerns of Reviewer#1 below.

1. All constructs: Readers should not be forced to track down construct details in half a dozen papers. Please include details of exactly where all Gag fusion constructs are linked to their tags.

Response: We apologize for the lack of information regarding Gag fusion constructs. We have now added this information to our revised text (**Supplementary Table 2**).

2. Figure 1: The authors should include a table of all Gag-interacting proteins identified by MS and their spectral count numbers, as well as what proteins were identified in control TAP tag pulldowns. This will be of interest to readers.

Response: We apologize for the inaccurate description in our proteomic analysis. Indeed, we performed an In-Gel tryptic Digestion (IGD) of silver-stained bands excised from Gag-TAP-associated protein samples under the strict condition in which the TAP-only lane had no observable band. In accordance with the reviewer's request, we have added detailed information on the proteins identified from each band in **New Supplementary Table 1**. We also more precisely describe the procedure for IGD and peptide mapping analysis in the **New Figure 1a** and the Materials and Methods section (**Page 19, Line 2**).

3-1. Figure 2: Did the authors use full-length APC fusions? If not, please explain.

Response: We did not use the full-length APC protein since it was quite difficult to synthesize in a wheat cell-free system due to its larger size (> 300 kD). We explain this issue in our revised text (**Page 6, Line 4**).

3-2. Also, based on the cartoon in panel A, it looks as though the GST tag is fused to the N-terminus of MA, which might be expected to perturb its folding. Please address this issue.

Response: We appreciate the reviewer's insight. Indeed, we used Gag proteins fused with GST at the N-terminus of MA in our binding experiments. As per the reviewer's suggestion, we newly created all Gag proteins with a C-terminal GST-tag and repeated the experiments with these constructs. We obtained very similar results to those using N-terminal tagged Gag proteins. These new data have been provided in our revised Figures (**New Figure 2 and New Figure 4b**).

4. Figure 3: In panels a-d please show the virion immunoblots and include the Gag CA (p24), p41 and Pr55 bands on the blots. This is necessary to validate release values. Also, with the SW480 blot (panel b), the cellular Gag bands for the APC- lane are much less bright than for the APC+ lanes, which could readily explain for the lack of Gag protein release as virions. Please use a better gel. I would also note that the top of the APC- blot in panel b looks highly contrasted. Perhaps a better blot here would be in order too.

Response:

1) We fully agree with this request and accordingly have included full size immunoblot figures that include the Pr55, p41 and p24 bands (**New Figure 3**).

2) We apologize for the poor quality of the SW480 blot. **New Figure 3b** now includes a higher quality image.

5-1) Figure 4 (= New Figure 5): Panel A: I do not see a significant difference between the examples of diffuse and PM staining. Better examples should be used. The authors also should use less subjective measurements of diffuse, PM and PM+Int staining.

Response: We apologize for the poor quality of the figure in question. We have improved it in the revised paper (**New Figure 5a**). Quantitative analysis of subcellular localization of Gag-GFP is rather well established and has already been used in many other studies (e.g. Neil et al, PLoS Pathogens, 2006; Miyakawa et al, PLoS Pathogens, 2009). Briefly, 20 random fields of HeLa cells expressing HIV-1 Gag-GFP protein were inspected, and we counted the numbers of cells in which Gag-GFP was observed as punctate fluorescence at the plasma membrane (PM) only, as diffuse cytoplasmic fluorescence only, or at intracellular sites as well as at the PM. Enumeration of cells with each pattern of Gag-GFP localization was expressed as a percentage the total of the cells counted. The data we provide are representative of three separate experiments. We explain this in more detail in the Materials and Methods, citing the aforementioned references, in our revised text (**Page 22, Line 8**).

5-2) Panel B: I don't find the FRET data convincing.

Response: We acknowledge this concern. In general, for most cell biological applications, FRET exhibits a low signal to noise ratio due to the fluorescent properties of the YFP/CFP tags. To more precisely monitor intracellular Gag-Gag interactions, we conducted a recently developed NanoBRET (nano-bioluminescence resonance energy transfer)-based protein-protein interaction assay, which uses NanoLuc Luciferase as the BRET energy donor and HaloTag protein labeled with the NanoBRET 618 fluorophore, as the energy acceptor to measure the interaction of specific protein pairs (Machleidt et al., ACS Chem Biol., 2015). We found that the depletion of APC indeed reduced the NanoBRET signal based on the Gag-Gag interaction. These new data are now shown in **New Fig. 5b** and the procedures are described in the Materials and Methods in our revised text (**Page 23, Line 13**).

5-3) *Panel C: Please show the entire virion blot.*

Response: As suggested, we now include full size virion immunoblots (**New Figure 5c**).

5-4) *Panel D: Did the authors score for p24 levels at later timepoints? If so, what were the results?*

Response: We have added data for “15 days post infection” in the same experiment (**New Figure 5d**).

6. *Figure 5 (= New Figure 6): Assuming that "APC targets cellular mRNA to actin-rich plasma membrane protrusions" (page 9), the vRNA localization is likely to be independent of the cis-active vRNA control elements. A control 24xMS2 RNA expressed from a non-retroviral plasmid ought to be tested. It also would be useful to graph cumulative RNA (MS2-GFP) line plots (summed from 20 or so images of each type) for quantification purposes.*

Response:

1) We fully accept this as a valid comment. In accordance with the reviewer's suggestion, we have generated a non-retroviral plasmid encoding firefly luciferase fused with 24xMS2-binding sites (pLuc-MSx24) (Jang et al., RNA Biol., 2016) and investigated its subcellular localization. We did not see any differences in its localization even in the presence or absence of APC expression, suggesting that APC does not affect the non-retroviral RNA with MS2-binding sites. These new data have been added as **New Supplementary Figure 3**.

2) Due to differences of the sizes and shapes of the cells, we could not overlay the line

plot data derived from different cells. Instead, we added another representative example of cell images (**Supplementary Figure 2c**). We also provide a bar chart indicating the percentage of cells with vRNA at the PM ($n > 20$) (**New Figure 6e**).

7. *Figure 6-7: I think the data from these VS model are not so convincing, and that the figures and their discussion can be deleted.*

Response: We agree that removing Figure 6 would not substantially diminish the impact of our study. However, we feel that Figure 7 might provide insights into the cell-to-cell spread of HIV-1, which we do consider to be an important point in our study. Therefore, we have improved Figure 7 in our revised manuscript. Please also see our response to Reviewer #2 on this issue below.

8. *Supplementary Figure 3: An analyses of Fyn(10)WT, Fyn(10)6A2T and Fyn(10)RKswitch does contribute to the manuscript, but instead of using the alphascreen, and colocalization studies, the authors should simply perform parallel studies to Figure 3a and include the results as a figure in the paper itself.*

Response: We fully accept this as a valid comment. In accordance with the reviewer's suggestion, the data in original Sup Fig 3 are now shown in **New Figure 4**. Moreover, we performed a HIV-1 production assay and AlphaScreen assay using WT, 6A2T, and RKswitch mutants (**New Figure 4a, 4b**). The results of these assays indicated that APC could interact with WT and RKswitch (both of which have highly basic residues in Gag), resulting in the enhancement of HIV-1 production. These results are in line with our conclusions in the original version of the manuscript.

9. *Discussion: Assuming Figures 6-7 are deleted, the Discussion should be re-written to reflect this.*

Response: We have modified the Discussion accordingly.

Responses to the comments of Reviewer #2

We sincerely appreciate the helpful and constructive suggestions by this reviewer. Reviewer#2 also regards this as an interesting study and has indicated that our current data define the role of APC as an important new host factor participating in the polarized localization of viral assembly to the site of cell-cell contact during virological synapse formation. This reviewer regards our experiments to have been well performed

and is supportive of our overall hypothesis. However, this referee does have concerns that epithelial cell lines were predominantly used in our study and has suggested that the manuscript would be strengthened by experiments that show whether APC is expressed in primary T cells, and whether a knock down or knock out of APC in primary T cells has an effect on HIV-1 infection and cell-to-cell spread. We have responded to the concerns of Reviewer #2 below.

1) Would be important to evaluate the levels of endogenous APC in primary T cells and determine whether this interaction occurs within infected primary T cells.

Response: We accept this as a valid point. Accordingly, we confirmed the substantial expression of endogenous APC in primary CD4⁺ T cells (**New Figure 5e**). We also confirmed the pivotal function of endogenous APC in HIV-1 replication using primary CD4⁺ T cells via an shRNA-mediated knockdown experiment (**New Figure 5e**). To examine the endogenous APC-Gag interaction in HIV-infected T cells, we performed immunoprecipitation analysis using a latently HIV-1 infected CD4⁺ T cell line, ACH2, following treatment with PMA to activate virus expression. Consequently, we found that HIV-1 Gag was immunoprecipitated with endogenous APC in the CD4⁺ T cells (**New Figure 1e**). Moreover, we performed a cell-to-cell virus transfer experiment using HIV-infected primary CD4⁺ T cells as donor cells and LuSIV cells as the acceptor reporter cells for the detection and quantitation of HIV-1 replication (Roos et al., Virology, 2000) (**New Figure 7f**). Consequently, we found that targeted depletion of APC in the HIV-infected primary CD4⁺ T cells prominently reduced the cell-to-cell virus transfer. These new data together support our principal contention that APC promotes virus assembly and spread in CD4⁺ T cells.

2. Fig. 7: It is not clear whether the analysis is rigorously excluding cell-cell doublets. The study may also benefit from making stable cell lines that are expressing shRNA or have a deleted APC gene with CRISPR. It would be convincing to show that the levels of APC protein are lower in the presence of siAPC vs siCtrl.

Response: We acknowledge the reviewer's concern here. We excluded cell-cell doublets by detecting disproportions between cell size vs. cell signal in order to analyze the exact efficiency of viral transfer. Our new flow cytometry data are now presented in **New Figure 7d and 7e**.

In accordance with the reviewer's request, we performed a cell-to-cell viral transfer assay using primary human CD4⁺ T cells stably expressing APC-targeted shRNA via a lentivirus vector. We confirmed that the shRNA-mediated APC knockdown in T cells significantly decreases the cell-to-cell viral transfer. We also found that the level of APC protein is prominently lower in the presence of shAPC as compared with shCtrl cells.

These new data are included as **New Figure 7f**.

Responses to the comments of Reviewer #3

We deeply appreciate the careful analysis and constructive suggestions made by this reviewer. This reviewer was convinced by our experimental strategy and results but had concerns relating to the potential physiological relevance of Wnt signaling in APC-mediate HIV-1 assembly. As requested, we have revised the manuscript to address these concerns as indicated below.

1) It is formally possible that effects of APC gain or loss of function experiments on HIV assembly are due to APC's role in Wnt signaling, i.e. that they are mediated by b-catenin dependent signaling rather than or in addition to its interaction with gag. In such a scenario activation of Wnt signaling by APC knockdown or in APC mutated colorectal cancer cells would prevent HIV assembly while inhibition of wnt signaling by full-size APC would stimulate it. It should be tested whether simultaneous knockdown of beta-catenin or expression of dominant-negative TCF blocks the consequences of APC knockdown in some of the key experiments shown in Figs. 4-7. Similarly, effects of b-catenin transfection or treatment of cells with Wnt3a should be determined.

Response: We acknowledge these valid suggestions. Accordingly, we performed several new experiments to investigate whether Wnt/ β -catenin signaling affects the function of APC on HIV-1 assembly and production.

1) We found that the expression of beta-catenin in 293T cells had no observable effects on the APC-mediated enhancement of HIV-1 production, whereas it could significantly induce TCF4-dependent transcriptional activity (**Supplementary Figure 1a**).

2) We also found that the expression of dominant-negative TCF4 in SW480 cells (APC-mutated colorectal cancer cells) had no significant effects on HIV-1 production but reduces TCF4-dependent transcriptional activity (**Supplementary Figure 1b**).

From these results, we conclude that Wnt signaling does not directly contribute to the role of APC in HIV-1 production.

2 Please explain briefly what the pNL4-3 vector encodes for.

Response: We apologize for not explaining this abbreviation. pNL4-3 vector encodes full-length HIV-1 genome derived from HIV-1_{NL4-3} molecular clone obtained from the NIH AIDS Reagent Program. Upon transfection, this clone directed the production of infectious virus particles in a wide variety of cells. We have added this information to **Supplementary Table 1**.

3 What is the purpose of the Fyn part in fig. 3d?

Response: In the absence of Gag MA domain, which is essential for Gag membrane binding, HIV-1 cannot bud from cells. Meanwhile, the addition of N-terminal 10-amino-acid sequence of Fyn kinase [Fyn(10)], which serves as a heterologous membrane binding signal, is sufficient to partially rescue membrane binding of MA-deficient Gag and hence the budding of MA-deficient virus (Chukkapalli et al., J Virol, 2008). To address whether the MA domain is important for the functional interaction of Gag with APC in the absence of the membrane binding defect, we used MA-deficient Gag conjugated with the Fyn sequence. We have addressed these issues briefly in our revised text (**Page 7, Line 10**).

4. Fig. 4c and d would fit better with Figure 3.

Response: We have added APC-knockdown experiments using primary CD4+ T cells, and confirmed that the endogenous APC plays role in the function in HIV-1 spread (**New Figure 5e**).

5. Fig. 6 This a collection of neat experiments. However, how can authors state that GAG, vRNA and APC are enriched in protrusion if one doesn't see the rest of the cell? Along that line, is it possible that absence of Gag and vRNA at protrusion after APC knockdown simply reflects lack of membrane association as shown for vRNA in fig. 5?

Response: We appreciate that this is a valid concern. As suggested by both reviewer #1 and the editor, we have excluded the original Figure 6 (trans-well assay data) from our revised manuscript. Since APC is predominantly localized at the cell protrusion or cell-cell contact sites rather than the rest of the cell body, we believe that APC is important for the accumulation of Gag and vRNA at such cell protrusion sites to enhance the efficiency of viral release/spread. These data will be further explored in our future studies.

Concerns of the Editor

Furthermore, there is an inconsistency in Figure 7a and 7b that needs to be addressed. The y-axis labeling suggests that “relative amount of Gag/vRNA at cell-to-cell contact site” is shown, while the figure legends suggests that “percentage of cells that Gag/vRNA was found at the cell-cell contact site” is shown.

Response: We apologize for this inconsistency and have amended our description in the

figure appropriately (**New Figure 7c**).

Other issue

- 1) We have newly added Masaki Anraku as a co-author of this paper since he conducted the FACS analyses to rigorously exclude cell-cell doublets in the cell-to-cell virus transfer assay. We confirmed that all co-authors have admitted it.
- 2) We found that the sequence of Fyn(10)6A2T shown in Old Supplementary Figure 3 contained a minor error, which originated from the error in previous paper by Llewellyn et al (J Virol 87, 6441, 2013). The correct 6A2T sequence is AWEAIALAPGGAAQYTLT (the positions of Thr are different), which is shown correctly in the supplementary figure in a former paper by Chukkapalli et al (PNAS 107, 1600, 2010). We have corrected this accordingly in New Figure 4a in our revised manuscript. This should not change any conclusion in the manuscript of our previous papers, but we are very sorry for the confusion that we made.
- 3) Words of the title were changed from “via interacting” to “via interaction” in order to make it clear.

REVIEWERS' COMMENTS:

Reviewer #1 (Remarks to the Author):

On the whole, the revisions incorporated into the Miyakawa et al. manuscript have improved the paper so that it should be of interest to Nature Communications readers. Although in some cases the effects of APC perturbation on virus production appears to be small, the authors in general have done a good job at quantification. I have only a few minor points of concern:

1. Supplementary Table 1: The authors show only single proteins identified for each of the cut out bands. I find it surprising that additional proteins were not identified in the MS sequencing. If other proteins were identified, even if their spectral count numbers were less than those of the proteins listed, they should be included in the table.
2. Supplementary Table 2: I still find the information on the constructs inadequate. In a supplement, it should not be difficult to show cartoon maps of the plasmids (which the authors presumably have), and indicate exactly what the junction sequences are.
3. Figures 5d-e: Full size Gag blots of the 15 day harvests similar to that shown in Figure 5c would be welcome.
4. Figures 5a and 6d: The analyses of these fluorescent images still borders on subjective, making the conclusions based on a very small number of images, or on subjective criteria as to the definitions of image classes.

Reviewer #2 (Remarks to the Author):

The revised manuscript by Miyakawa shows new data in primary T cells that knocking down APC reduced the efficiency of cell-cell transmission. While the magnitude of the influence of knockdown on viral transmission is modest, the data overall remains supportive of the role that APC is important in virus assembly and cell-cell transmission of HIV. Also improved in this manuscript is the presentation of the flow cytometry data that addressed issues of doublet discrimination. Overall the manuscript is improved, appears to be technically sound and with new data adequately addressed the concerns of this reviewer.

Reviewer #3 (Remarks to the Author):

The major concern of this reviewer was that APC's effect on viral budding might be related to its role in wnt signaling rather than or in addition to its binding to gag. The authors show that neither activation of the pathway by b-catenin nor inhibition by dnTCF affects viral budding excluding such a scenario. All other issues were also addressed appropriately so that there are no further remaining criticisms.

As a minor correction the legend to Supplementary Fig. S1c,d should be changed from "Note that the expression of dnTCF4 in SW480 cells (APC-mutated colorectal cancer cells) had no observable effects on the APC-mediated enhancement of HIV-1 production, whereas

it significantly reduced TCF4-dependent transcriptional activity.”

to

“Note that the expression of dnTCF4 in SW480 cells (APC-mutated colorectal cancer cells) had no observable effects on HIV-1 production, whereas it significantly reduced TCF4-dependent transcriptional activity.” because the APC-mediated enhancement of HIV production was not studied in conjunction with dnTCF in this experiment.

Responses to the comments of Reviewer #1

On the whole, the revisions incorporated into the Miyakawa et al. manuscript have improved the paper so that it should be of interest to Nature Communications readers. Although in some cases the effects of APC perturbation on virus production appears to be small, the authors in general have done a good job at quantification. I have only a few minor points of concern:

Response: We sincerely appreciate the helpful and constructive suggestions by this reviewer. We have addressed the concerns raised by this reviewer as follows:

1. Supplementary Table 1: The authors show only single proteins identified for each of the cut out bands. I find it surprising that additional proteins were not identified in the MS sequencing. If other proteins were identified, even if their spectral count numbers were less than those of the proteins listed, they should be included in the table.

Response: According to the reviewer's request, we have included several other proteins with lower spectral count numbers in **New Supplementary Table 1**.

2. Supplementary Table 2: I still find the information on the constructs inadequate. In a supplement, it should not be difficult to show cartoon maps of the plasmids (which the authors presumably have), and indicate exactly what the junction sequences are.

Response: This is a valid comment for readers. We have added the cartoon map with junctional linker sequences of newly created plasmids (**New Supplementary Table 2**).

3. Figures 5d-e: Full size Gag blots of the 15 day harvests similar to that shown in Figure 5c would be welcome.

Response: As suggested, we have added the full size Gag blots in **New Figure 5d-e**.

4. Figures 5a and 6d: The analyses of these fluorescent images still borders on subjective, making the conclusions based on a very small number of images, or on subjective criteria as to the definitions of image classes.

Response: We analyzed the subcellular localization of Gag-GFP at least 100 randomly selected cells. Also, we analyzed vRNA localization at least 20 cells in randomly selected fields in confocal images. We added the detailed descriptions in our revised text (**Page 9 Line 3; Page 11, Line 1**)

Responses to the comments of Reviewer #2

The revised manuscript by Miyakawa shows new data in primary T cells that knocking down APC reduced the efficiency of cell-cell transmission. While the magnitude of the influence of knockdown on viral transmission is modest, the data overall remains supportive of the role that APC is important in virus assembly and cell-cell transmission of HIV. Also improved in this manuscript is the presentation of the flow cytometry data that addressed issues of doublet discrimination. Overall the manuscript is improved, appears to be technically sound and with new data adequately addressed the concerns of this reviewer.

Response: We wish to express our highest appreciation to Reviewer#2 for his/her insightful comments, which have helped us significantly improve our manuscript.

Responses to the comments of Reviewer #3

The major concern of this reviewer was that APC's effect on viral budding might be related to its role in wnt signaling rather than or in addition to its binding to gag. The authors show that neither activation of the pathway by b-catenin nor inhibition by dnTCF affects viral budding excluding such a scenario. All other issues were also addressed appropriately so that there are no further remaining criticisms.

As a minor correction the legend to Supplementary Fig. S1c,d should be changed from "Note that the expression of dnTCF4 in SW480 cells (APC-mutated colorectal cancer cells) had no observable effects on the APC-mediated enhancement of HIV-1 production, whereas it significantly reduced TCF4-dependent transcriptional activity." to "Note that the expression of dnTCF4 in SW480 cells (APC-mutated colorectal cancer cells) had no observable effects on HIV-1 production, whereas it significantly reduced TCF4-dependent transcriptional activity." because the APC-mediated enhancement of HIV production was not studied in conjunction with dnTCF in this experiment.

Response: We appreciate the valuable comments from Reviewer#3 and believe that our manuscript has been improved by his/her constructive suggestions. We acknowledge these valid suggestions regarding the figure legend of Fig. S1c,d. We have modified the legend according to the reviewer's indication.